# Molecular screening and dynamics simulation reveal potential phytocompounds in *Swertia chirayita* targeting the UspA1 protein of *Moraxella catarrhalis* for COPD therapy

**Md. Moin Uddin[1], Md. Shydhur Rahman Chowdhury[2], Md. Arju Hossain[3], Asif Ahsan[1], Md. Tanvir Hossain[1], Abdul Barik[1], Md. Arif Hossen[4], Md. Faisal Amin[5], Rafsan Abir[6], Mohammad Shah Alam[7], Md Habibur Rahman[8]\*, M. Nazmul Hoque[9]\***

1 Department of Biotechnology, Bangladesh Agricultural University, Mymensingh, Bangladesh, 2 Department of Pharmacy, Mawlana Bhashani Science and Technology University, Tangail, Bangladesh, 3 Department of Biochemistry and Biotechnology, Khwaja Yunus Ali University, Sirajganj, Bangladesh, 4 Department of Biochemistry and Molecular Biology, Mawlana Bhashani Science and Technology University, Tangail, Bangladesh, 5 Department of Biochemistry and Molecular Biology, The University of Texas Rio Grande Valley, Edinburg, Texas, United States of America, 6 Department of Microbiology, Primeasia University, Dhaka, Bangladesh, 7 Department of Anatomy and Histology, Gazipur Agricultural University, Gazipur, Bangladesh, 8 Center for Advanced Bioinformatics and Artificial Intelligence Research, Department of Computer Science and Engineering, Islamic University, Kushtia, Bangladesh, 9 Molecular Biology and Bioinformatics Laboratory, Department of Gynecology, Obstetrics and Reproductive Health, Gazipur Agricultural University, Gazipur, Bangladesh

\* habib@iu.ac.bd (MHR); nazmul90@bsmrau.edu.bd (MNH)

## Abstract

Chronic obstructive pulmonary disease (COPD) is a global health burden, with *Moraxella catarrhalis* significantly contributing to acute exacerbations and increased healthcare challenges. This study aimed to identify potential drug candidates in *Swertia chirayita*, a traditional Himalayan medicinal plant, demonstrating efficacy against the ubiquitous surface protein A1 (UspA1) of *M. catarrhalis* through an *in-silico* computational approach. The three-dimensional structures of 46 phytocompounds of *S. chirayita* were retrieved from the IMPPAT 2.0 database. The structures underwent thorough analysis and screening, emphasizing key factors such as binding energy, molecular docking performance, drug-likeness, and toxicity prediction to assess their therapeutic potential. Considering the spectrometry, pharmacokinetic properties, docking results, drug likeliness, and toxicological effects, five phytocompounds such as beta-amyrin, calendol, episwertenol, kairatenol and swertanone were identified as the inhibitors of the UspA1 in *M. catarrhalis*. UspA1 demonstrated binding affinities of −9.1 kcal/mol for beta-amyrin, −8.9 kcal/mol for calendol, −9.4 kcal/mol for episwertenol, −9.6 kcal/mol for kairatenol, and −9.0 kcal/mol for swertanone. All of these affinities were stronger than that of the control drug ceftobiprole, which had a binding score of −6.6 kcal/mol. The toxicity analysis confirmed that all five compounds are safe potential therapeutic options, showing no toxicity or carcinogenicity. We also performed a 100 ns molecular dynamics simulation of the phytocompounds to analyze their stability

**Data availability statement:** All relevant data are within the manuscript. The article contains the data utilized to support the results of the in-silico study.

**Funding:** The author(s) received no specific funding for this work.

**Competing interests:** The authors have declared that no competing interests exist.

and interactions as protein-ligand complexes. Among the five screened phytocompounds, beta-amyrin and episwertenol exhibited favorable characteristics, including stable root mean square deviation values, minimal root mean square fluctuations, and consistent radius of gyration values. Throughout the simulations, intermolecular interactions such as hydrogen bonds and hydrophobic contacts were maintained. Additionally, the compounds demonstrated strong affinity, as indicated by negative binding free energy values. Taken together, findings of this study strongly suggest that beta-amyrin and episwertenol have the potential to act as inhibitors against the UspA1 protein of *M. catarrhalis*, offering promising prospects for the treatment and management of COPD.

## Introduction

COPD is a serious respiratory condition encompassing chronic bronchitis and emphysema, impacting millions of people globally [1–3]. It is the third leading cause of death worldwide and thus is a global public health concern [4]. Bacterial infections play a pivotal role in exacerbating COPD symptoms, with *M. catarrhalis*, *Streptococcus pneumoniae*, *Haemophilus influenzae*, and *Pseudomonas aeruginosa* being the most common pathogens [1,2,5]. Among these, *M. catarrhalis* is notably implicated in acute exacerbations and long-term disease progression. It colonizes the respiratory tract, particularly targeting small airway epithelial cells (SAECs) through its UspA1 protein, which facilitates adhesion, immune evasion, and sustained inflammation. This interaction contributes to airway obstruction, chronic inflammation, and tissue damage [5,6]. In COPD, persistent inflammation weakens the immune system, impairing the respiratory defense against infections and worsening disease progression. Immune dysregulation in COPD involves chronic inflammation, impaired immune responses, and exaggerated innate immunity, leading to tissue damage and increased susceptibility to infections. This dysfunction in immune cells and cytokine imbalance plays a crucial role in COPD pathogenesis and exacerbations [7,8]. This chronic inflammatory state disrupts the lung microenvironment and microbiome, further exacerbating inflammation. Such dysregulation not only accelerates COPD progression but also increases the risk of developing lung cancer due to a pro-carcinogenic environment [7,9].

Numerous studies have established a correlation between *M. catarrhalis* colonization in the respiratory tract and the progression or exacerbation of COPD [5,10]. *M. catarrhalis*, a gram-negative diplococcus bacterium, develops otitis media in newborns and spreads to the airways; it is typically detected in the sputum of people with COPD [11,12]. It is the second most frequent cause of exacerbated symptoms in humans with COPD, leading to severe diseases and fatalities [13]. Membrane components ubiquitous UspA1 protein, expressed by *M. catarrhalis*, plays a significant role in COPD pathogenesis. It promotes bacterial adhesion to airway epithelial cells, evades immune responses, and induces inflammation and disease progression [6,14]. Additionally, UspA1 interacts with host proteins such as carcinoembryonic antigen-related cell adhesion molecule 1 (CEACAM1) and extracellular matrix proteins, enhancing bacterial colonization and persistence in the lungs, further worsening chronic inflammation and tissue damage [15]. Furthermore, earlier studies demonstrated that when *M. catarrhalis* interacts with epithelial cells that possess the CEACAM1 cytoplasmic domain and the UspA1 outer membrane protein, it induces apoptosis in SAECs [16,17]. Targeting UspA1 could offer a therapeutic strategy for managing COPD exacerbations linked to *M. catarrhalis* infections. Currently, antibiotics play a crucial role in preventing COPD exacerbations. Antimicrobial resistance (AMR) exacerbates challenges in managing *M. catarrhalis* infections [18,19], as the bacterium frequently produces β-lactamase enzymes that render it

resistant to common antibiotics such as penicillin and amoxicillin. While prophylactic antibiotics can reduce the frequency of exacerbations, long-term use carries the risk of side effects and promotes the development of bacterial resistance [20,21]. This necessitates exploration of alternative therapeutic agents, particularly phytochemicals with proven antimicrobial activity.

Recently, research into the use of natural plants for alleviating major human diseases has gained momentum. Traditional medicine, especially medicinal plants, remains the most affordable and accessible treatment in primary healthcare, offering potential remedies for a wide range of conditions [22–25]. *S. chirayita*, a medicinal herb from the Gentianaceae family [26], has been traditionally used for treating respiratory and systemic ailments. Its bioactive constituents, including xanthones, flavonoids, and secoiridoid glycosides, exhibit potent antimicrobial, anti-inflammatory, and antioxidant properties [27]. While direct evidence against *M. catarrhalis* is limited, studies indicate that the plant's extracts demonstrate efficacy in inhibiting microbial growth and warrants further investigation to establish a direct link to its therapeutic use in COPD management. For effective primary health care, patients must have access to and utilize appropriate drugs. While extensive studies on the AMR and multidrug resistance (MDR) of *M. catarrhalis* exist [21,28,29], there is a lack of research focusing on effective phytocompounds targeting the UspA1 protein. Recent *in-silico* studies have identified promising candidates, such as beta-amyrin and episwertenol, from *S. chirayita* have demonstrated stable binding interactions with the UspA1 protein of *M. catarrhalis* during molecular docking and dynamics simulations. Such findings highlight their potential to disrupt bacterial adhesion and immune evasion, reducing inflammation and exacerbations in COPD patients. Thus, this research aimed to identify natural drug candidates from *S. chirayita* that are effective against the UspA1 protein of *M. catarrhalis* in COPD patients through computational methods. Molecular docking is extensively employed to analyze protein structure-activity relationships, assisting in identifying novel UspA1 inhibitors among phytocompounds [30]. It is crucial for predicting how small ligand molecules interact with specific binding sites on target proteins. Additionally, molecular dynamics simulations help analyze the conformational changes during protein-ligand interactions. Further experimental validation is essential to translate these computational insights into therapeutic applications for *M. catarrhalis*-induced COPD.

## Materials and methods

This study employed a comprehensive *in-silico* approach to identify potential therapeutic candidates from *S. chirayita* for *M. catarrhalis*-induced COPD, focusing on UspA1 and CEACAM1 as critical targets. Protein structures were retrieved from the Protein Data Bank (PDB) or modeled using homology techniques when required. The structures were preprocessed by removing water molecules, adding polar hydrogens, and optimizing them for docking. Phytocompounds from *S. chirayita* were sourced from established phytochemical databases and prepared by energy minimization. Protein-ligand docking was performed using AutoDock or similar tools to identify compounds with high binding affinity, emphasizing interactions with key active site residues. Molecular dynamics simulations (MDS) were then conducted over 100 ns to evaluate the stability and dynamics of these interactions, capturing key parameters like RMSD, RMSF, and binding free energy. Finally, MM/GBSA analysis provided insights into the binding energetics, enabling a detailed evaluation of potential drug candidates Fig 1.

### Ethical statement

This *in-silico* computational study did not involve human subjects and therefore did not require ethical approval. Furthermore, the authors declare that this manuscript, submitted to

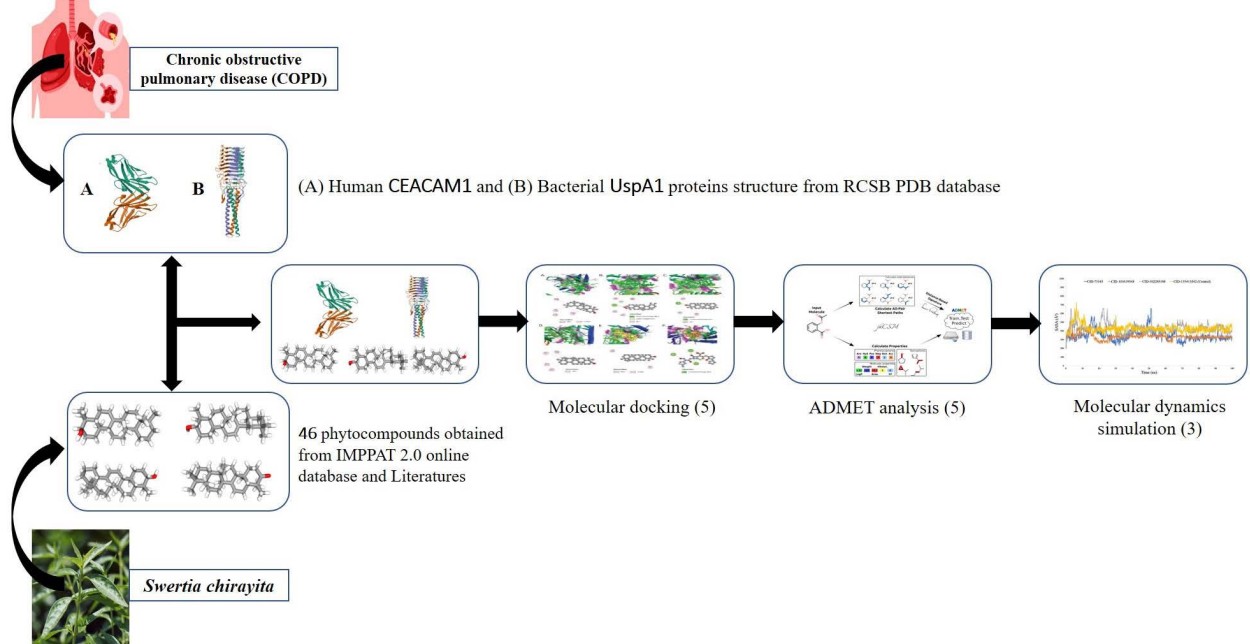

**Fig 1. Schematic representation on *in-silico* identification of potential drug candidates from *S. chirayita* targeting the UspA1 protein of *M. catarrhalis* and CEACAM1 receptor of human associated with COPD.** A total of 46 phytocompounds were retrieved from the IMPPAT 2.0 database and analyzed through molecular docking, revealing five effective inhibitors viz. beta-amyrin, calendol, episwertenol, kairatenol, and swertanone against UspA1. Drug likeliness and toxicity assessments confirmed their safety, and 100 ns molecular dynamics simulations demonstrated the stability and favorable interactions of beta-amyrin and kairatenol with UspA1, suggesting their potential as therapeutic options for *M. catarrhalis* related COPD.

PLOS ONE, has been prepared with full adherence to responsible research practices and in accordance with the guidelines of publication ethics.

## Protein preparation

The three-dimensional (3D) structure of UspA1 (PDB ID: 3NTN, resolution 2.20 Å and 220 amino acid sequences) and CEACAM1 (PDB ID: 6XNW, resolution 1.90 Å and 108 amino acid sequences) were retrieved from the RCSB Protein Data Bank (RCSB PDB; accessed: June 10, 2024). PyMOL v 2.3.3 [31] and BIOVIA Discovery Studio Visualizer v4.5 [32] (accessed: June 10, 2024) were utilized to prepare the proteins by eliminating hetero atoms, water molecules, and unnecessary residue connections [33]. Energy minimization using Swiss-PDB Viewer v4.1.0 [34] was performed to stabilize the protein structures by resolving unfavorable contacts. Optimized proteins were then used for molecular docking analysis with UspA1 of *M. catarrhalis* and human CEACAM1 to evaluate the binding interactions of *S. chirayita* phytocompounds.

## Retrieval and preparation of phytocompounds from *S. chirayita*

Forty-six (n = 46) phytocompounds (ligands) of *S. chirayita* were retrieved in structured data file (SDF) format and canonical smiles from the Indian Medicinal Plants, Phytochemistry And Therapeutics 2.0 (IMPPAT 2.0) database [35] (accessed: June 12, 2024). The IMPPAT database encompasses 4,010 Indian medicinal plants, 17,967 phytochemicals, and 1,095 therapeutic applications. Ceftobiprole, a broad-spectrum fifth-generation cephalosporin antibiotic, was

selected as the control drug due to its demonstrated efficacy in inhibiting *M. catarrhalis*, a key pathogen implicated in COPD-related infections [31,32]. Using PyMOL [36], we converted all ligand files from SDF to PDB format. We employed an online SMILES translator and structure file generator (https://cactus.nci.nih.gov/translate/) to obtain the 3D structures of the selected ligands (accessed: June 12, 2024). Subsequently, each molecule was prepared and utilized as a ligand in the molecular docking studies with the target protein.

## Virtual screening and molecular docking of the phytocompounds

Utilizing AutoDock Vina v1.2.1 [37], molecular docking was carried out. It includes a scoring function, multi-ligand docking, and batch mode for large-scale docking. AutoDock Vina is open access software for molecular docking and virtual screening. The pdbqt file format was utilized to forecast the binding affinity of protein and ligand molecules and involve their interactions [38]. Using AutoDock Vina, the 3D formats of UspA1 and CEACAM1, along with their corresponding ligands, were generated in pdbqt format by eliminating heteroatoms, water molecules, and cofactors. A central grid box was used by AutoDock Vina to designate the binding site. By optimizing the diameters (X, Y, Z), the whole region of each protein and ligand binding site was covered. The grid boxes were set up with the following dimensions (measured in 'Å'): X = 14.346, Y = −83.336, and Z = 7.376 for the 3NTN structure, and X = −34.460, Y = −16.803, and Z = −42.872 for the 6XNW structure. Due to the utilization of extensive grid boxes, the default value of 8 for the "exhaustiveness" parameter in AutoDock Vina was maintained. A molecular docking analysis was also conducted using the PyRx virtual screening software (v1.0) [39] combined with the AutoDock Vina platform. This analysis aimed to evaluate the binding scores of the target proteins with the selected phytochemicals. Default settings and parameters were used in PyRx analysis.

## Protein-phytocompounds interactions analysis

We employed the PyMOL [36] and BIOVIA Discovery Studio Visualizer v4.5 [33] to examine or visualize the binding orientation of the protein-phytocompounds (ligand) complexes. It created two-dimensional models of protein-ligand interactions and graphs demonstrating electrostatic, hydrophobic, and hydrogen bonds. The interaction between phytocompounds and the active sites of UspA1 and CEACAM1 proteins was better understood with the use of these illustrations, present in *M. catarrhalis* and humans, respectively, thus affecting their affinity levels.

## Prediction of pharmacokinetics and drug likeliness properties of the phytocompounds

Following the docking and protein-ligand interactions analyses, the *in-silico* computational pharmacokinetics (PK) approach determined the temporal dynamics of drug absorption, distribution, metabolism, excretion, and toxicity (ADMET) of the phytocompounds. We evaluated the PK and ADME characteristics of the selected phytocompounds using SwissADME web tool [40] (accessed: June 20, 2024). SwissADME helps filter and optimize lead compounds, ensuring they exhibit favorable pharmacokinetic and toxicological profiles before advancing to more extensive experimental evaluations [40]. Among them, we examined several parameters, including the count of rotatable bonds (nRB), topological surface area (TPSA), the number of hydrogen bond acceptors (HBAs) and hydrogen bond donors (HBDs), molecular mass (MM), and any violations of Lipinski's Rule of Five [40,37]. In addition to assessing the ADME profile, it is important to evaluate the potential toxicity of chemical compounds that may harm human organs or cells. To accomplish this, we utilized

the pkCSM online server (accessed: June 20, 2024), which predicts various toxicity endpoints, enhancing our understanding of the safety profile of the compounds before proceeding to further experimental stages [41]. Furthermore, admetSAR v1.0 [38] and Pro Tox-II [39] web servers (accessed: June 22, 2024), were used to gauge the toxicity level of the chosen phytocompounds.

## High-affinity docking with known antibiotics

We conducted further docking analysis of well-known and approved antibiotics, including grepafloxacin, cefditoren, clavulanic acid, loracarbef, ceftriaxone, tetracycline, roxithromycin, cefdinir, gemifloxacin, sparfloxacin, clarithromycin, cefuroxime, besifloxacin, and dirithromycin. These antibiotics are effective in treating *M. catarrhalis*-mediated diseases and other bacterial infections [32]. This analysis evaluated the interactions of these antibiotics with the UspA1 and CEACAM1 proteins to identify suitable control drugs. We utilized the DrugBank online database (accessed: July 10, 2024) to identify the most similar antibiotics for our analysis, and examined their highest affinity [42]. To facilitate the docking analysis, we obtained drugs in PDB format from the PubChem database (accessed: July 10, 2024) approved for treating bacterial diseases.

## Validation of UspA1 protein structure by Ramachandran plot

To validate the structural quality of the UspA1 protein, a Ramachandran plot analysis was conducted using tools such as PROCHECK [43]. This analysis calculated the phi ($\phi$) and psi ($\psi$) dihedral angles of the protein backbone, classifying residues into core, allowed, and disallowed regions based on stereochemical constraints. If over 90% of residues were located in the favored and allowed regions, indicating high structural reliability and suitability for downstream computational studies. The Ramachandran plot was generated to visually assess the conformation and ensure minimal outliers [34], with findings integrated into the validation section for robust analysis.

## Molecular dynamics simulation of the phytocompounds and UspA1 protein

Based on the molecular screening, pharmacokinetics and drug likeliness properties, top three phytocompounds such as beta-amyrin (CID: 73145), episwertenol (CID: 101619548), and kairatenol (CID: 102285188) were selected for molecular dynamics simulation (MDS) along with the control drug, ceftobiprole (CID: 135413542). To validate the reliability of the protein-ligand complex structures, we performed molecular MDS for approximately 100 nanoseconds (ns). The "Desmond v3.6 Program" from Schrodinger (https://www.schrodinger.com/ac) was used to model the molecular dynamics of the top-docked protein-ligand complex structures in a Linux infrastructure [44]. A basic TIP3P water model was employed to establish a linear volume within a simple orthorhombic periodic container, confined to a 10 Å region. After constructing the water model using the "System Builder" tool, neutralizing sodium and chloride ions were added to balance the system. Energy minimization was performed using the OPLS3 force field to optimize the structure. Subsequently, the systems were simulated for 100 nanoseconds (ns) under NPT conditions at a temperature of 300 K and atmospheric pressure (1.01325 bar) [45]. During this analysis, 50 ps-grabbing pauses were conducted simultaneously, demonstrating a binding potency of 1.2 kcal/mol. Several parameters were utilized to assess the equilibrium of the protein-ligand complex, including root-mean-square deviation (RMSD), protein-ligand contacts (P-L), intramolecular hydrogen bonding, solvent accessible surface area (SASA), radius of gyration (Rg), molecular surface

area (MolSA), and polar surface area (PSA) values [41]. These parameters provided a comprehensive evaluation of the stability and interactions within the complex throughout the simulation [41]. The MDS screenshots were generated and captured using Schrodinger's Maestro application (v 9.5). Analysis of the simulation event and evaluation of the MD simulation's reliability were conducted utilizing the interaction diagram derived from the Desmond modules within the Schrodinger suite [25,41].

## Post MDS thermal MM-GBSA analysis

We further conducted molecular mechanics-generalized born surface area (MM-GBSA) calculations to estimate the free energy of the complex during the 100 ns simulation, utilizing the thermal_mmgbsa.py Python module. Subsequently, the Desmond molecular dynamics trajectory was segmented into 20 discrete snapshots, and MM-GBSA analysis was performed on each snapshot to isolate the interactions between the ligand and receptor. This approach allowed us to comprehensively assess the binding affinity and dynamics of the ligand-receptor complex over the simulation period, providing valuable insights into their interactions [44].

## Results

### Binding interaction of the phytocompounds of *S. chirayita*

This research aimed to utilize *in-silico* computational techniques to identify potential drug candidates in *S. chirayita*, a widely used medicinal plant native to the temperate Himalayan region. These drug candidates were specifically targeted against the UspA1 protein of *M. catarrhalis*, which plays a key role in COPD infections. Additionally, the phytocompounds of *S. chirayita* were also screened for their interactions with the human CEACAM1 protein which plays important roles in immune regulation, cell adhesion, and inflammation. Ceftobiprole was selected as the control drug for comparison. We also evaluated the molecular docking scores of UspA1 and CEACAM1 proteins against various drugs approved for treating bacterial diseases of humans, including COPD. Table 1 presents the interactions between the phytocompounds with high molecular screening (docking) scores and the protein structures of UspA1 (PDB ID: 3NTN) and CEACAM1 (PDB ID: 6XNW). Molecules with a more negative binding affinity were deemed to have the most favorable interactions. Based on molecular screening, the five top candidates selected for further investigation were beta-amyrin (CID: 73145), calendol (CID: 604983), episwertenol (CID: 101619548), kairatenol (CID: 102285188), and swertanone (CID: 102285187). The UspA1 protein of *M. catarrhalis* showed higher binding scores (e.g., −9.1 kcal/mol for beta-amyrin, −8.9 kcal/mol for calendol, −9.4 kcal/mol for episwertenol, −9.6 kcal/mol for kairatenol, and −9.0 kcal/mol for swertanone) compared to the human CEACAM1 protein. However, these phytocompounds displayed binding energies of −6.2, −7.3, −5.8, −6.4, and −6.8 kcal/mol for CEACAM1, respectively. Moreover, we investigated the docking score of UspA1 with the control drug ceftobiprole, revealing a value of −6.6 kcal/mol (Table 1). Ceftobiprole demonstrated significant hydrogen bonding interactions with key residues in UspA1 receptor, including Asp355, Glu357, Gly359, and Asn366 (Table 1), supporting its utility as a benchmark. The interactions of beta-amyrin, calendol, episwertenol, kairatenol, and swertanone with the specific molecular regions within the binding pockets of both UspA1 and CEACAM1 proteins are illustrated in Fig 2 and S1 Fig, respectively. In the UspA1 structure, Leu[360] was found to be a common binding site for ceftobiprole, episwertenol, and kairatenol. Similarly, Met[7], Pro[8], and Lys[15] were found to be shared interacting residues for calendol, episwertenol, and kairatenol. Through molecular screening of UspA1 with approved drugs, we identified several compounds exhibiting promising binding affinities. These include grepafloxacin (−6.5 kcal/mol), cefditoren (−6.8 kcal/mol), clavulanic acid

**Table 1. Interaction of phytocompounds showing high docking scores for UspA1 (PDB ID: 3NTN) of *M. catarrhalis* and CEACAM1 (PDB ID: 6XNW) of human protein structures.**

| Receptors (PDB ID) | Ligands (CID) | Binding scores (kcal/mol) | Interacting amino acids | |
|---|---|---|---|---|
| | | | Hydrogen bonds | Hydrophobic interactions |
| UspA1 (3NTN) | Ceftobiprole (135413542; control drug) | −6.6 | Asp[355], Glu[357], Gly[359], Asn[366] | Leu[360], Leu[361] |
| | Beta-amyrin (73145) | −9.1 | NA | Lys[367], His[368], His[369], His[370] |
| | Calendol (604983) | −8.9 | NA | Lys[352], Leu[354] |
| | Episwertenol (101619548) | −9.4 | NA | Leu[360] |
| | Kairatenol (102285188) | −9.6 | Glu[357] | Leu[360], Leu[361] |
| | Swertanone (102285187) | −9.0 | NA | Leu[361], Gly[362] |
| CEACAM1 (6XNW) | Ceftobiprole (135413542; control drug) | −5.7 | Arg[38], Ala[39], Thr[101] | NA |
| | Beta-amyrin (73145) | −6.2 | NA | Leu[24], Leu[28], Tyr[31], Ile[50] |
| | Calendol (604983) | −7.3 | Glu[5], Ser[6] | Met[7], Pro[8], Ala[12], Lys[15] |
| | Episwertenol (101619548) | −5.8 | Ala[12] | Met[7], Pro[8], Lys[15] |
| | Kairatenol (102285188) | −6.4 | NA | Met[7], Pro[8], Ala[12], Lys[15] |
| | Swertanone (102285187) | −6.8 | NA | Pro[8], Tyr[107] |

NA: Not applicable/found.

(−5.7 kcal/mol), loracarbef (−6.6 kcal/mol), ceftriaxone (−7.4 kcal/mol), tetracycline (−7.5 kcal/mol), roxithromycin (−5.6 kcal/mol), cefdinir (−6.7 kcal/mol), gemifloxacin (−6.1 kcal/mol), sparfloxacin (−7.4 kcal/mol), clarithromycin (−6.2 kcal/mol), cefuroxime (−6.5 kcal/mol), besifloxacin (−6.9 kcal/mol), and dirithromycin (−6.4 kcal/mol). The binding scores of CEACAM1 against grepafloxacin, cefditoren, clavulanic acid, loracarbef, ceftriaxone, tetracycline, roxithromycin, cefdinir, gemifloxacin, sparfloxacin, clarithromycin, cefuroxime, besifloxacin, and dirithromycin were −5.1, −5.4, −4.3, −5.3, −6.3, −6.3, −5.0, −5.5, −4.7, −5.6, −5.5, −5.4, −5.4, and −4.7 kcal/mol (Table 2). Thus, UspA1 protein exhibited a significantly higher binding affinity for the tested approved drugs than the CEACAM1 protein. This finding suggested that the molecular interactions between UspA1 and tested drugs are more favorable, potentially indicating that UspA1 may be a more viable target for therapeutic intervention in conditions related to *M. catarrhalis*.

## Pharmacokinetics and toxicological properties of the phytochemicals

The pharmacokinetics (PKs) characteristics and ADME properties of five selected phytocompounds along with control are shown in Table 3. PKs analysis of the phytocompounds included factors such as intestinal and colon epithelial cancer cell line (humans) absorption, blood-brain barrier (BBB), central nervous system (CNS) permeability, volume of distribution at steady-state (VDss), cytochrome P450 (CYP) inhibitors, and total clearance (Table 3). Each compound exhibited a human intestinal absorption rate exceeding 93%, while the permeability values of Caco-2 varied between 1.219 and 1.325 cm/s. The projected level of VD in L/Kg was between 0.205 and 0.316 L/Kg, which is excessively low, indicating higher distribution of the compounds in plasma. Skin permeability values ranging from −2.726 to −2.834 cm/s were recorded for all substances included in this analysis, suggesting that the skin is not quite permeable. Furthermore, we found that each of the five

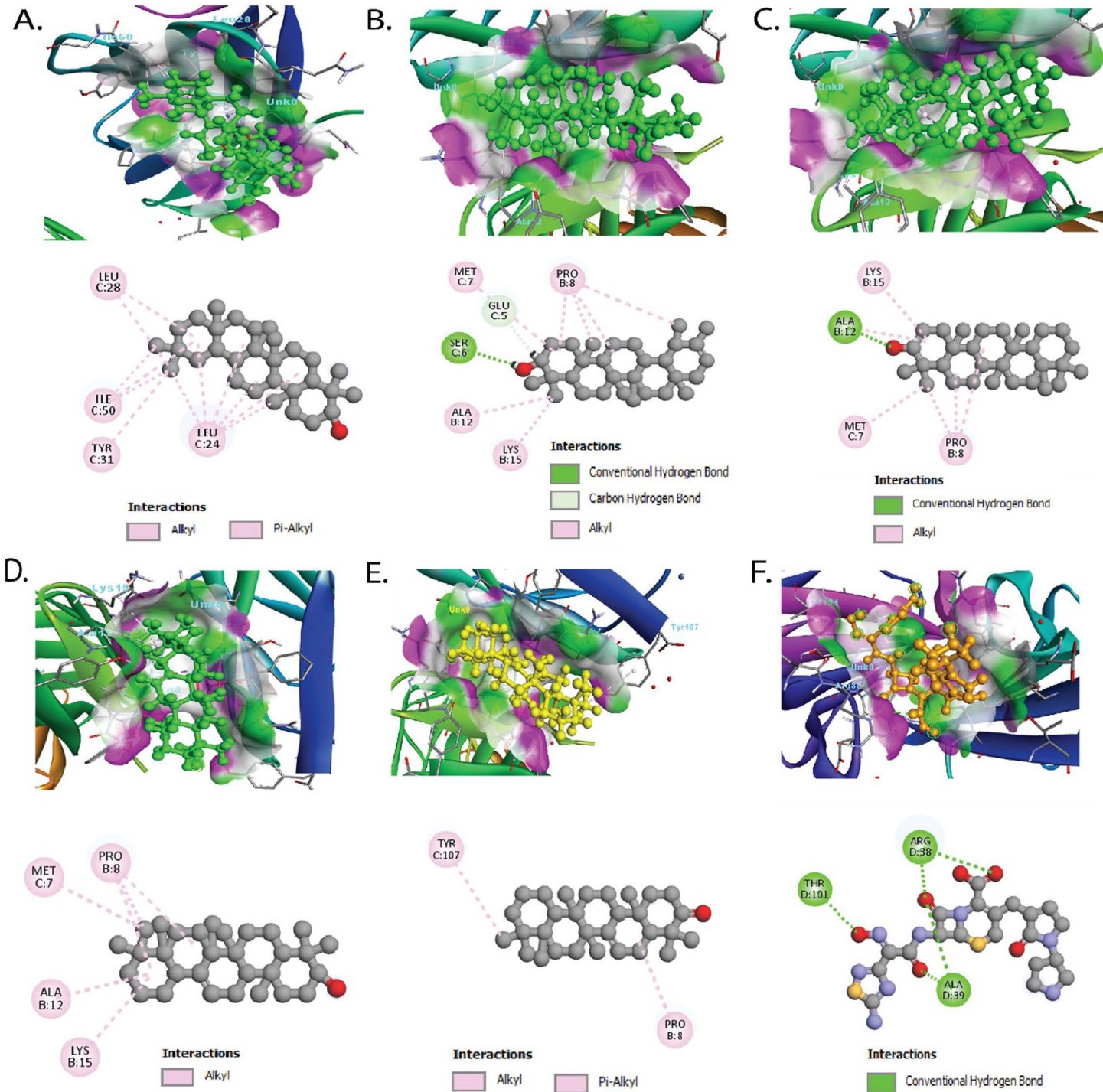

**Fig 2. The interaction profile of the docking complex reveals the protein-ligands (phytocompounds) interactions.** On upper panel, (A) UspA1 and beta-amyrin, (B) UspA1 and calendol, (C) UspA1 and episwertenol, (D) UspA1 and kairatenol, and (E) UspA1 and swertanone. The lower panel shows the three-dimensional (3D) structure of all complexes.

**Table 2. Binding scores of target proteins UspA1 of *M. catarrhalis* and CEACAM1 of humans to various known antibiotics.**

| Generic name | DrugBank Accession Number | Generation | Binding Score of UspA1 (kcal/mol) | Binding Score of CEACAM1 (kcal/mol) |
|---|---|---|---|---|
| Grepafloxacin (CID: 72474) | DB00365 | 4th generation | −6.5 | −5.1 |
| Cefditoren (CID: 9870843) | DB01066 | 3rd generation | −6.8 | −5.4 |
| Clavulanic acid (CID: 5280980) | DB00766 | 3rd generation | −5.7 | −4.3 |
| Loracarbef (CID: 5284585) | DB00447 | 2nd generation | −6.6 | −5.3 |
| Ceftriaxone (CID: 5479530) | DB01212 | 3rd generation | −7.4 | −6.3 |
| Tetracycline (CID: 54675776) | DB00759 | 3rd generation | −7.5 | −6.3 |
| Roxithromycin (CID: 6915744) | DB00778 | 2nd generation | −5.6 | −5.0 |
| Cefdinir (CID: 6915944) | DB00535 | 3rd generation | −6.7 | −5.5 |
| Gemifloxacin (CID: 9571107) | DB01155 | 4th generation | −6.1 | −4.7 |
| Sparfloxacin (CID: 60464) | DB01208 | 3rd generation | −7.4 | −5.6 |
| Clarithromycin (CID: 84029) | DB01211 | 2nd generation | −6.2 | −5.5 |
| Cefuroxime (CID: 5479529) | DB01112 | 2nd generation | −6.5 | −5.4 |
| Besifloxacin (CID: 10178705) | DB06771 | 4th generation | −6.9 | −5.4 |
| Dirithromycin (CID: 6473883) | DB00954 | 4th generation | −6.4 | −4.7 |

**Table 3. Pharmacokinetic profile of the selected phytocompounds of *S. chirayita*, considering their absorption, distribution, metabolism, and excretion (ADME) properties.**

| Properties | Model Name | Phytocompounds | | | | | |
|---|---|---|---|---|---|---|---|
| | | Ceftobiprole (control) | Beta-amyrin | Calendol | Episwertenol | Kairatenol | Swertanone |
| Absorption | Intestinal absorption (human) | 28.469 | 93.733 | 94.584 | 94.703 | 93.962 | 97.463 |
| | Skin permeability | −2.735 | −2.811 | −2.834 | −2.798 | −2.811 | −2.726 |
| | Caco-2 permeability | −0.293 | 1.226 | 1.223 | 1.219 | 1.221 | 1.325 |
| Distribution | VDss (human) | −0.5 | 0.268 | 0.316 | 0.236 | 0.259 | 0.205 |
| | Fraction unbound (human) | 0.735 | 0 | 0 | 0 | 0 | 0 |
| | BBB permeability | −1.389 | 0.667 | 0.661 | 0.675 | 0.67 | 0.692 |
| | CNS permeability | −3.656 | −1.773 | −1.891 | −1.898 | −1.901 | −1.726 |
| Metabolism | CYP1A2 inhibitor | No | No | No | No | No | No |
| | CYP2C19 inhibitor | No | No | No | No | No | No |
| | CYP2C9 inhibitor | No | No | No | No | No | No |
| | CYP2D6 inhibitor | No | No | No | No | No | No |
| | CYP3A4 inhibitor | No | No | No | No | No | No |
| Excretion | Total clearance | 1.26 | −0.044 | −0.023 | 0.081 | 0.081 | 0.029 |
| | Skin sensitization | No | No | No | No | No | No |

Here: Caco-2: human colon epithelial cancer cell line, VDss: volume of distribution at steady-state, BBB: blood-brain barrier, CNS: central nervous system, CYP: cytochrome P450.

phytocompound has the potential to penetrate the BBB and CNS (Table 3). Additionally, we discovered that the chosen phytocompounds may all be quickly digested by the liver, lowering their therapeutic concentration in the body. Additional physicochemical properties of the selected phytocompounds, including H-bond acceptors, H-bond donors, TPSA (Å²), and molar refractivity as outlined in Table 4, indicate their potential as drug candidates. The selected phytocompounds also possessed molecular weights below 500 g/mol (beta-amyrin: 426.72 g/mol, episwertenol: 426.72 g/mol, calendol: 426.72g/mol, kairatenol:

**Table 4. Physicochemical properties, lipophilicity, water solubility, and drug-likeness properties of the selected five phytocompounds along with the control drug.**

| Properties | Model name | Compounds | | | | | |
|---|---|---|---|---|---|---|---|
| | | Ceftobiprole (control) | Beta-amyrin | Calendol | Episwertenol | Kairatenol | Swertanone |
| Physicochemical | Molecular weight | 534.57 g/mol | 426.72 g/mol | 426.72 g/mol | 426.72 g/mol | 426.72 g/mol | 424.70 g/mol |
| | No. of H-bond acceptors | 10 | 1 | 1 | 1 | 1 | 1 |
| | No. of H-bond donors | 5 | 1 | 1 | 1 | 1 | 0 |
| | Molar refractivity | 139.11 | 134.88 | 135.14 | 134.88 | 134.88 | 133.92 |
| | TPSA (Å²) | 256.98 | 20.23 | 20.23 | 20.23 | 20.23 | 17.07 |
| Lipophilicity | Consensus Log Po/w | −1.25 | 7.18 | 7.03 | 7.17 | 7.16 | 7.20 |
| Water solubility | Log S (ESOL) | −1.26 | −8.25 | −8.09 | −8.25 | −8.25 | −8.04 |
| | Solubility class | Very soluble | Poorly soluble | Poorly soluble | Poorly soluble | Poorly soluble | Poorly soluble |
| Drug-likeness | Lipinski violation | No; 2 violations: MW > 500, NorO > 10 | Yes; 1 violation: MLOGP > 4.15 | Yes; 1 violation: MLOGP > 4.15 | Yes; 1 violation: MLOGP > 4.15 | Yes; 1 violation: MLOGP > 4.15 | Yes; 1 violation: MLOGP > 4.15 |
| | Bioavailability score | 0.17 | 0.55 | 0.55 | 0.55 | 0.55 | 0.55 |
| Medicinal chemistry | PAINs | 0 alert | 0 alert | 0 alert | 0 alert | 0 alert | 0 alert |

426.72 g/mol, and swertanone: 424.70g/mol). Notably, each phytocompound complies with the guidelines of Lipinski's "rule of five." The water solubility properties showed that all reported substances were poorly soluble in water.

We also predicted the toxicological properties of the five phytocompounds along with the control drug, and the results indicated that AMES was not affected by the investigated phytocompounds (i.e., not mutagenic) (Table 5). However, all the phytocompounds exhibited minimal inhibition of the hERG gene, which is crucial for cardiac health. These phytocompounds are all considered to be of "class III" toxicity based on the expected acute oral toxicity values. Given their potential for drug development, five phytocompounds selected in this study are generally considered acceptable, as they exhibit $LD_{50}$ values below 5000 mg/kg. Specifically, episwertenol and swertanone exhibited $LD_{50}$ values below 5000 mg/kg, whereas beta-amyrin, calendol, chiratenol, and kairatenol displayed values exceeding 5000 mg/kg. Thus, most of our selected phytocompounds fall within the desirable range. Additionally, none of the chosen phytocompounds displayed cytotoxic effects, hepatotoxicity and carcinogenicity in particular (Table 5). As a result, each of the five phytocompounds demonstrated effectiveness and showed promise as potential therapeutics due to their favorable oral bioavailability and safety profiles.

## Stereochemical quality of the target protein

The Ramachandran plot assessed the stereochemical quality of the modeled UspA1 protein structure, classifying the φ (phi) and ψ (psi) dihedral angles of amino acid residues into favored, allowed, and outlier regions (Fig 3). This analysis showed 90.3% of residues in the favored region (red), 8.2% in the allowed regions (yellow), and only 1.5% in the outlier regions (white) (Fig 3). The predominance of residues in favorable conformational regions confirms the structural reliability necessary for precise ligand-binding analysis.

**Table 5. Toxicological properties of selected five compounds include drug-induced hERG inhibition, AMES toxicity, carcinogens, acute oral toxicity, acute toxicity (LD$_{50}$ in mg/kg), hepatoxicity of selected five compounds.**

| Compounds | hERG inhi2ition | RAT (LD$_{50}$) mg/kg | Class | AMES | Carcinogens | Acute oral toxicity | Hepatoxicity |
|---|---|---|---|---|---|---|---|
| Ceftobiprole (control) | Weak | 8000 | 6 | No | No | III | Active |
| Beta-amyrin | Weak | 70000 | 6 | No | No | III | Inactive |
| Calendol | Weak | 70000 | 6 | No | No | III | Inactive |
| Episwertenol | Weak | 5000 | 5 | No | No | III | Inactive |
| Kairatenol | Weak | 70000 | 6 | No | No | III | Inactive |
| Swertanone | Weak | 5000 | 5 | No | No | III | Inactive |

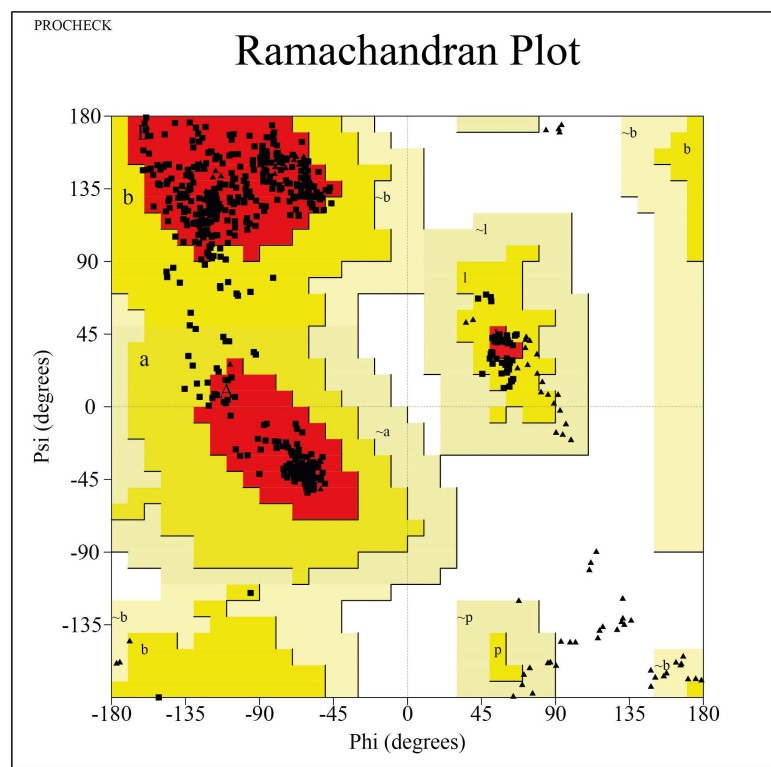

**Fig 3. A Ramachandran plot was constructed to illustrate the distribution of phi (φ) and psi (ψ) dihedral angles for the residues of UspA1.** The plot's color-coding highlights regions of conformational preference: red for favored core regions (α-helices and β-sheets), yellow for less favored but allowed regions, and white for outlier regions.

## Molecular dynamics simulation trajectories of the phytocompounds

While all five compounds showed promise as potential drug candidates, the top three such as beta-amyrin (CID: 73145), episwertenol (CID: 101619548), and kairatenol (CID: 102285188) along with the control drug, ceftobiprole (CID: 135413542), were selected for additional molecular screening due to their anticipated favorable pharmacokinetics and drug-likeness profiles. During molecular screening, the UspA1 protein of *M. catarrhalis* demonstrated higher binding stability compared to the human CEACAM1 receptor. Thus, we selected UspA1 protein for the MDS run due to its superior binding interaction and stability compared to the human CEACAM1 protein. Finally, beta-amyrin, episwertenol, and kairatenol,

in addition to the control drug ceftobiprole, were subjected to 100 ns of MDS for a more in-depth evaluation of their interactions and stability as drug candidates.

We calculated the RMSD to verify if the protein structure was stable and to find any orientation changes. For the more stable compounds, a lower RMSD value is indicative. RMSD values below 1.5 Å often indicate a more consistent docking, whereas values above 1.5 Å commonly represent average binding positions. Proteins with intrinsically flexible domains or loops may naturally exhibit higher RMSD values due to conformational adjustments that do not compromise overall binding stability. On the other hand, ligand binding can induce structural rearrangements in the protein, resulting in increased RMSD values that reflect dynamic adaptability rather than instability. The average change in RMSD for each protein-ligand interaction was more than sufficient, falling within an estimated range of 2–5 Å. During the 100 ns simulation trajectory, beta-amyrin (CID:73145) and kairatenol (CID:102285188) exhibited superior performance compared to the control and other compound, with RMSD values ranging from 3–5 Å. However, compound episwertenol (CID:101619548) demonstrated a maximal fluctuation in its value that exceeded what is typically deemed acceptable (Fig 4A). The RMSF plays a vital role in evaluating changes in specific regions of protein structures. Most of the changes, especially at the protein's ends, occur in its tails, which include the N- and C-terminal domains. Alpha helices and beta strands show lower fluctuation rates compared to loop regions due to their structural rigidity. RMSF aids in understanding protein dynamics comprehensively. This encompasses a range of 5 to 290 amino acid residues, where a lower value compared to a reference indicates lesser variation. Notably, during a 100 ns simulation period, the most stable fluctuations, ranging from 2 to 5 Å, were observed for 400 amino acid residues, specifically for beta-amyrin (CID:73145), kairatenol (CID:102285188), and control drug (CID:135413542), except for compound episwertenol (CID:101619548). Fluctuations moderately increased from 400 to 450 residues due to N and C terminal residues (Fig 4B). Calculating Rg is a crucial parameter to consider when predicting the structural behavior of a macromolecule, as it reflects changes in the overall compactness of the complex over time. The Rg readings for the phytocompound beta-amyrin (CID:73145), episwertenol (CID:101619548), kairatenol (CID:102285188), and the control drug ceftobiprole (CID:135413542) were observed to be 4.2, 4.3, 4.3, and 5.1 Å, respectively (Fig 5A). This indicates that the binding region of the protein remains relatively unchanged upon attachment of the selected ligand substances. Protein residues on their surfaces serve as reaction zones that either bond to other molecules or act as receptor ligands. This analysis offers valuable information regarding the molecule's solvation properties (hydrophilic or hydrophobic) and the kinetics of protein-ligand interactions. Fig 5B depicts the SASA measurement of protein molecules associated with each of the four ligand combinations. In this study, the SASA for the selected compounds and the control drug ranged from 198 to 724Å², indicating a significant level of interaction between the amino acids sequence and the chosen chemical through a combination of processes. A low SASA value signifies a tightly packed water-amino acid combination, whereas a high SASA value indicates a less stable structure with looser packing. The SASA values ranged from 198 to 655 Å² for all drugs except the control ceftobiprole (CID:135413542). A higher SASA value (ranging from 233 to 655 Å²) for CID-102285188 (kairatenol) suggested that many of the selected ligand molecules were found in complex systems involving an amino acid residue (Fig 5B). When using a probe diameter of 1.4 Å, the MolSA closely resembles a region defined by van der Waals surfaces. In this *in-silico* analysis, all of the three phytocompounds exhibited a distinct van der Waals interface space along with the control drug (Fig 6A). Furthermore, only molecules containing oxygen and nitrogen components demonstrated PSA. The selected phytocompounds beta-amyrin (CID:73145), episwertenol (CID:101619548), kairatenol (CID:102285188), and the control drug ceftobiprole (CID:135413542) exhibited notable PSA levels (Fig 6B).

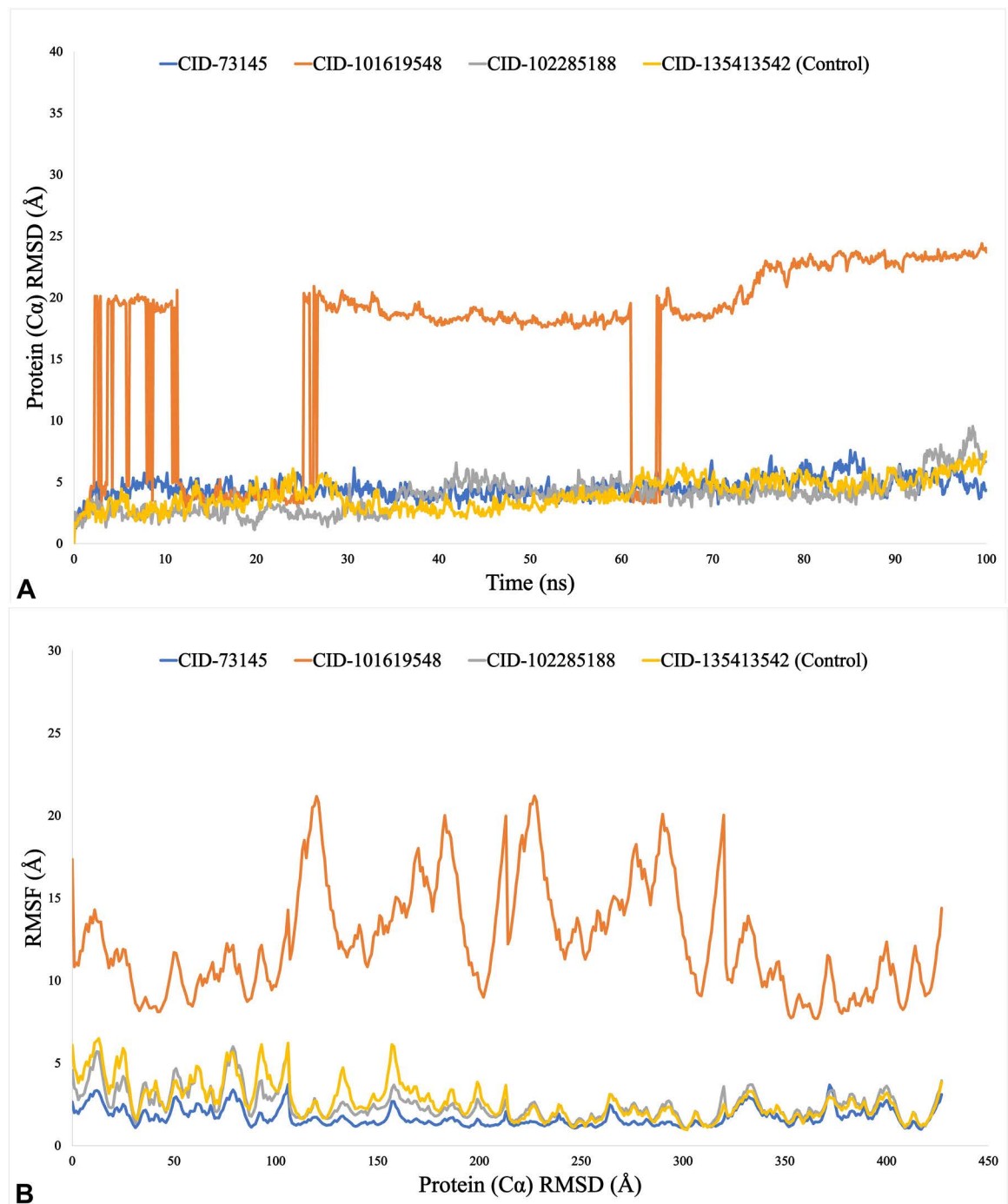

**Fig 4. The (A) root-mean-square deviation (RMSD) and (B) root-mean-square fluctuation (RMSF) values of the UspA1 protein (PDB ID: 3NTN), derived from the Cα atoms of the complex system with the three top phytocompounds and the control drug.** The selected phytocompounds, beta-amyrin (CID: 73145), episwertenol (CID: 101619548), kairatenol (CID: 102285188), and the control drug ceftobiprole (CID: 135413542), in complex with the protein, are represented by blue, orange, gray, and yellow, respectively.

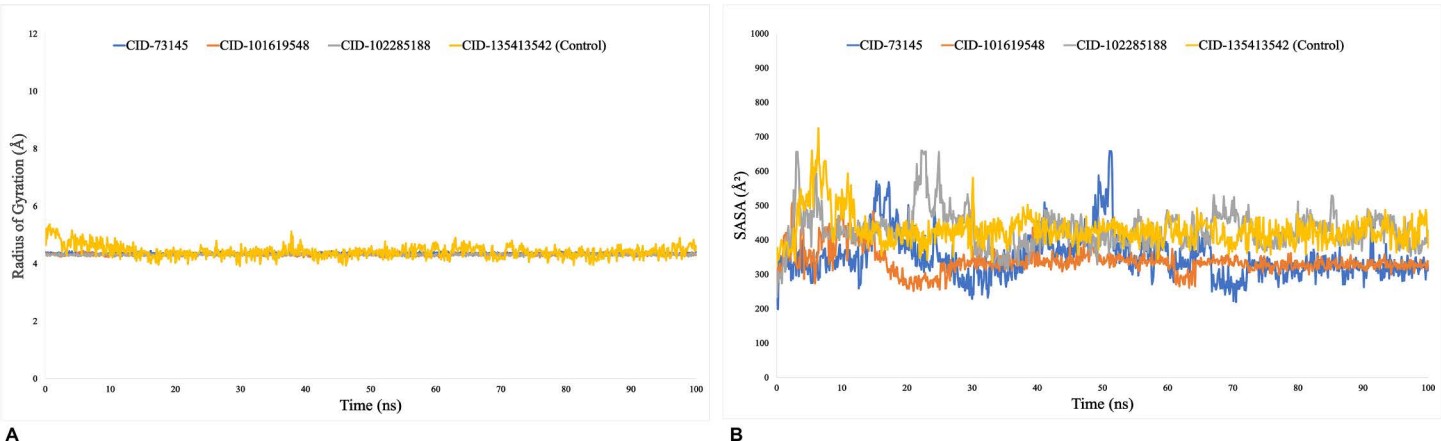

**Fig 5. The (A) radius of gyration (Rg) and solvent accessible surface area (SASA) values of the protein-ligand complexes measured at 100 ns of simulation.** The selected compounds, beta-amyrin (CID: 73145), episwertenol (CID: 101619548), kairatenol (CID: 102285188), and the control drug ceftobiprole (CID: 135413542), in complex with the protein, are represented by blue, orange, gray, and yellow, respectively.

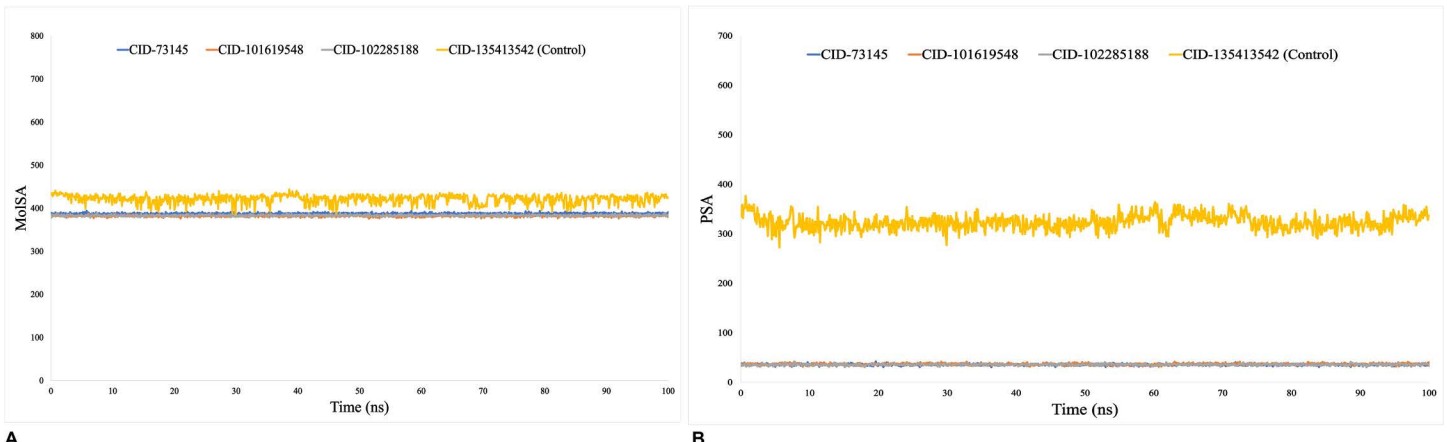

**Fig 6. The (A) molecular surface area (MolSA) and (B) polar surface area (PSA) values of the protein-ligand complexes measured at 100 ns of simulation.** The selected compounds, beta-amyrin (CID: 73145), episwertenol (CID: 101619548), kairatenol (CID: 102285188), and the control drug ceftobiprole (CID: 135413542), in complex with the protein, are represented by blue, orange, gray, and yellow, respectively.

During the 100 ns simulation, the "Simulation Interactions Diagram (SID)" was employed to examine intermolecular interactions between the selected phytocompounds and the control drug. The results indicated that all compounds consistently formed and maintained various interactions, including hydrophobic, ionic, water bridge, and hydrogen bonding, ensuring stable interactions with the target protein molecules (Fig 7). An increase in the number of hydrogen bonds between the target protein and its ligands indicates stable binding. Our analysis revealed that beta-amyrin (CID:73145; Fig 7A), episwertenol (CID:101619548; Fig 7B), kairatenol (CID:102285188; Fig 7C) and control drug (CID:135413542; Fig 7D) interacted through various bond types, including hydrophobic, ionic, and fluid connections within the protein structure. Throughout the entire 100 ns trajectories, each substance established numerous interactions that remained stable until the experiment's conclusion, fostering sustained interaction affinity with the selected proteins. All of the three phytocompounds

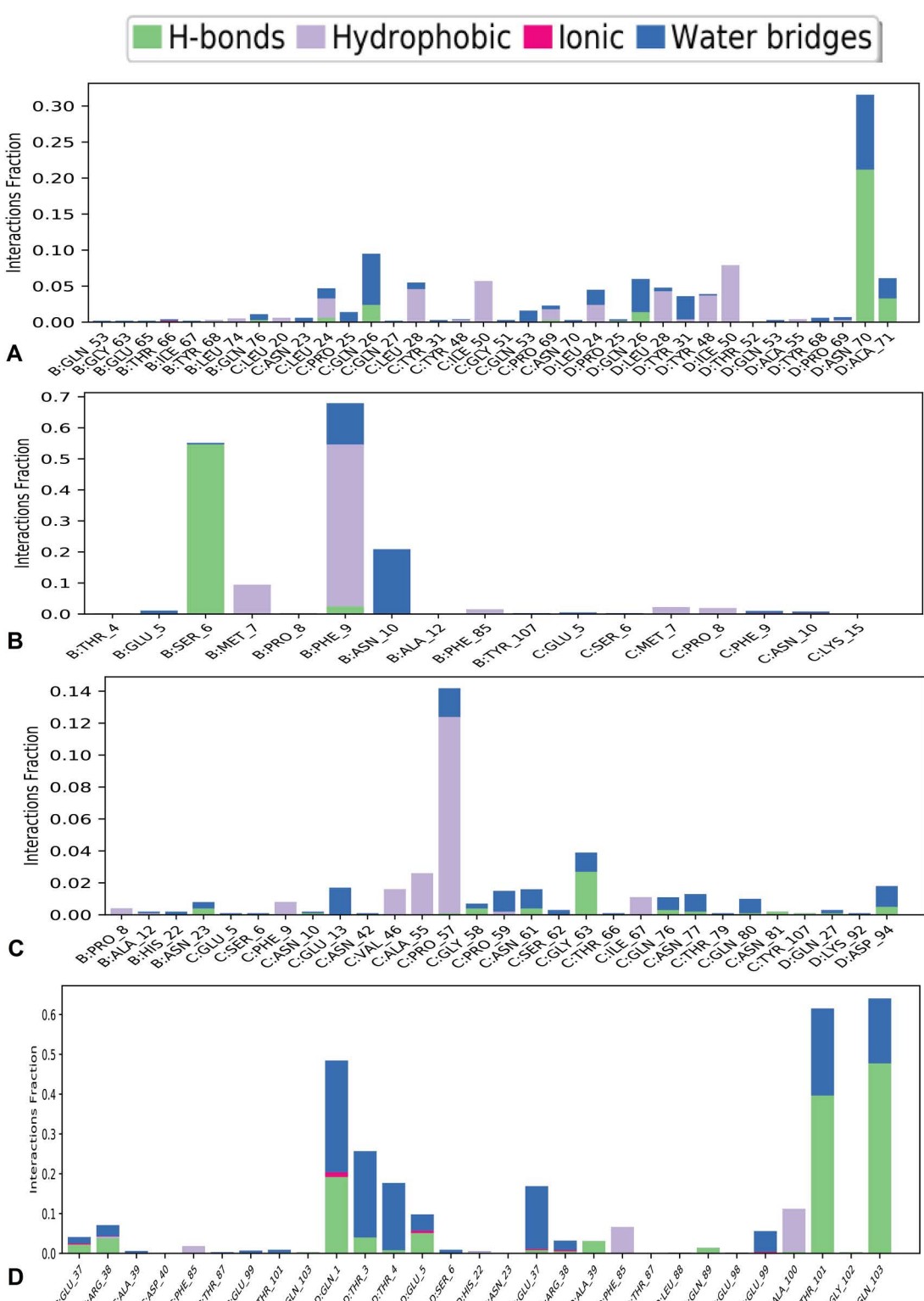

**Fig 7. The interactions between protein and ligands identified during the 100 ns simulation are shown in the stacked bar charts.** Herein, bar plots are showing intermolecular fractions for the compounds (A) beta-amyrin, (B) episwertenol, (C) kairatenol and (D) control drug ceftobiprole.

displayed commendable qualities, with beta-amyrin (Fig 7A) and episwertenol (Fig 7B) notably demonstrating an optimal level of MDS.

## Binding free energy calculation through MM/GBSA analysis

To evaluate the ligand-binding free energy to the protein receptor, MM/GBSA analysis was performed (Fig 8). Binding energy snapshots were captured every 20 ns from the 100 ns simulation trajectories. The key finding from this study was that each of the selected phytocompounds exhibited significantly higher thermal binding free energy compared to the control drug. Remarkably, our findings showed that beta-amyrin, episwertenol, and kairatenol exhibited thermal binding Gibbs free energy values of −54.21, −54.63, and −40.09 kcal/mol, respectively, highlighting their potential for stronger interactions compared to the control drug, which had a free binding energy of −42.033 kcal/mol. Taken together, these findings suggested that beta-amyrin and episwertenol had a greater potential for effective interaction with the target protein UspA1 (Fig 8).

## Discussion

The global prevalence of COPD is alarming, as it accounted for 3.23 million deaths (6% of total global mortality) in 2019, making it the third leading cause of death worldwide [46]. *M. catarrhalis* is an important pathogen found in the respiratory tract of individuals with COPD, particularly during acute exacerbations of the disease [18]. Targeting UspA1 in *M. catarrhalis* holds potential as a therapeutic approach in managing COPD, given its role in bacterial

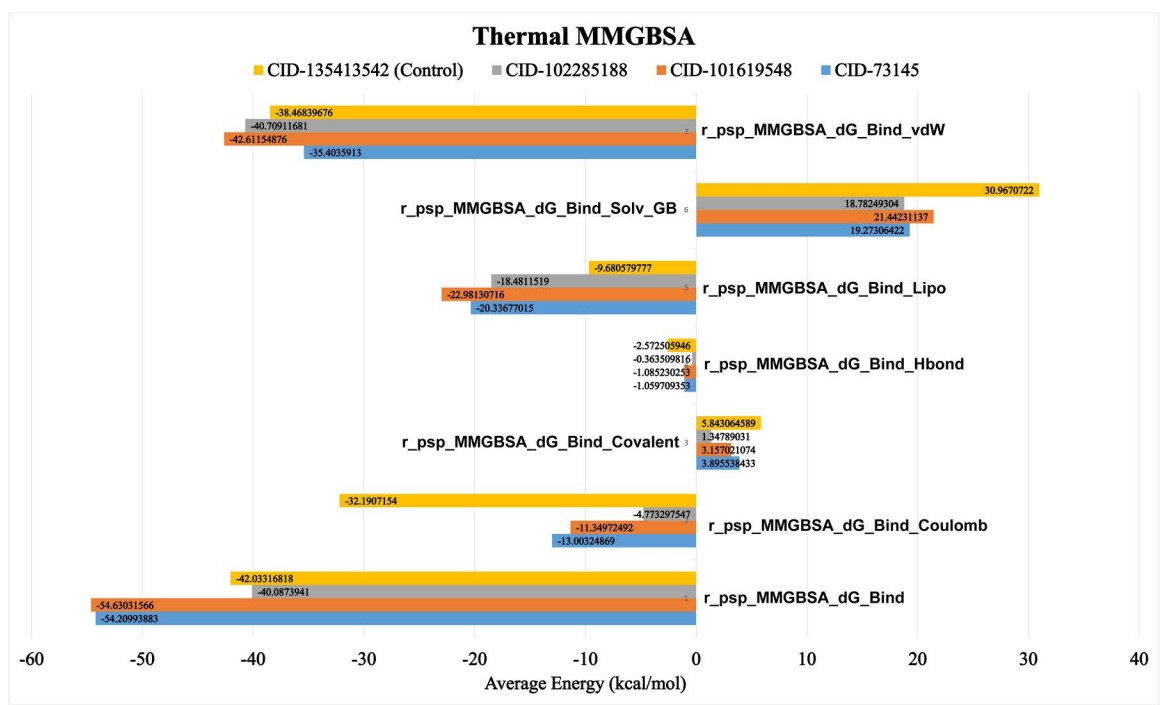

**Fig 8. Post-simulation thermal MM-GBSA.** The three top compounds CID: 73145 (beta-amyrin), CID: 101619548 (episwertenol), CID: 102285188 (kairatenol) and CID: 135413542 (control drug) containing the targeted protein UspA1were subjected to a post-simulation thermal MP-GBSA analysis. The cumulative MM-GBSA binding free energy was determined by contributions from Coulombic forces (Coulomb), covalent interactions (Covalent), hydrogen bonding (H-bond), lipophilic interactions (Lipo), generalized solvent effects, and van der Waals forces (VDW).

adhesion and immune modulation. UspA1 interacts with CEACAM1, a receptor on airway epithelial cells, which not only aids bacterial colonization but also enhances pro-inflammatory responses, contributing to the exacerbation of COPD [47]. On the other hand, CEACAM1 plays a key role in modulating immune responses by facilitating bacterial adhesion and regulating inflammatory processes in the lungs. In COPD, this receptor contributes to chronic inflammation and immune dysfunction [48]. Inhibiting UspA1 could reduce bacterial adherence and inflammation, potentially mitigating the chronic immune activation that worsens disease progression. The immune dysfunction observed in COPD, characterized by chronic inflammation and impaired defense against infections, could be modulated by disrupting the UspA1-CEACAM1 interaction. By preventing this interaction, it may restore more balanced immune responses, reducing the inflammatory burden and improving lung function [6]. Moreover, this strategy could offer an alternative to traditional antibiotics, particularly in light of rising antimicrobial resistance, by targeting the bacterial interaction mechanisms rather than relying on conventional drugs [42]. Thus, targeting UspA1 provides a dual benefit: alleviating bacterial-driven inflammation and potentially reducing the need for antibiotics in treating COPD exacerbations. This therapeutic approach could play a crucial role in improving the long-term management of COPD patients.

As drug resistance increases in conventional COPD treatments, the search for superior candidates intensifies, with natural plant-derived phytochemicals offering promising alternatives for COPD therapy. Therefore, we aimed to identify potential drug candidates from *S. chirayita* that effectively target the UspA1 (PDB ID: 3NTN) protein of *M. catarrhalis* through computational methods. We investigated the thermodynamic characteristics of five phytocompounds namely beta-amyrin, calendol, episwertenol, kairatenol, and swertanone, exhibited strong binding affinity to the UspA1 protein structure of *M. catarrhalis*. The UspA1 protein is integral to the ability of *M. catarrhalis* to cause persistent infections, evade the immune system, and contribute to disease progression, particularly in respiratory conditions like COPD [6,49]. Thermodynamic analysis showed that the studied compounds had lower hardness and higher softness values compared to the control drug, indicating greater reactivity. Our predicted target-ligand complex showed better binding scores than control ceftobiprole ($-6.6$ kcal/mol with UspA1 and $-5.7$ kcal/mol with CEACAM1), which might be identical for designing a drug. These results suggest improved thermodynamic properties, with enhanced potential for interactions with the target. Thermodynamic calculations predict reaction kinetics and molecular stability, where more negative values indicate a higher likelihood of binding with other molecules [50]. At the binding interface, hydrophobic interactions are crucial for ligand stabilization. Many hydrophobic atoms in the targeted molecules may facilitate the drug-target interaction [51]. Hydrogen bonds are vital when it comes to the specificity of ligand binding. Their significant work is directly integrated into GRID, a computational approach developed to determine energetically effective ligand binding sites on a selected protein molecule with a known structure. In molecular interaction analysis, the hydrophobic and hydrogen bonds are instrumental in designing a drug against harmful pathogens like *M. catarrhalis* [52].

Both UspA1 and CEACAM1 proteins had a promising binding affinity for fourteen approved drugs identified by the sequence interaction and docking studies. These drugs are commonly prescribed antibiotics that are effective against several bacterial diseases [32,46,53]. Patients with COPD have a few antibiotic options: tetracycline, trimethoprim/sulfamethoxazole, cephalosporins (cefuroxime, cefpodoxime, or cefdinir), ketolides (telithromycin), and advanced macrolides (azithromycin, clarithromycin). Some antibiotic treatments for complex patients include amoxicillin/clavulanic acid and respiratory fluoroquinolones such as moxifloxacin, gemifloxacin, and levofloxacin [54]. Our newly discovered drug candidates exhibited

better binding affinity than previously approved drugs, preventing COPD-related diseases (Tables 1 and 2). Subsequently, we evaluated the pharmacokinetic profiles of the chosen compounds. Optimizing pharmacokinetic properties is a critical step in determining whether a chemical has the efficiency to become a successful drug candidate [55]. This evaluation ensures that the compound possesses the necessary characteristics for absorption, distribution, metabolism, and excretion, allowing it to reach its target site effectively and safely [24,25]. Notably, all potential compounds exhibited TPSA values within the acceptable range, ranging from 17.07 to 40.46 Å$^2$. Molecular weight and TPSA are essential factors in determining the permeability of compounds, with lower values exhibiting enhanced permeability [56]. All the phytocompounds had molecular weights below 500 g/mol, which is a favorable indicator of their potential as drug candidates, aligning with the widely accepted rule of drug-likeness.

Another key finding of our study is that the drug-likeness and ADME properties of the studied compounds, along with the control, complied with Lipinski's "rule of five." All compounds met Lipinski's criteria, exhibiting lower molecular weights than the control. Furthermore, their physicochemical properties suggest promising potential as drug candidates. ADME predictions revealed that the docked phytocompounds fell within the acceptable range for drug development. However, since our target is brain disorders, these compounds must also demonstrate the ability to permeate the blood-brain barrier, a crucial consideration in drug development [57]. Furthermore, none of the studied drug candidates exhibited carcinogenic properties. An *in-silico* AMES test was performed to assess the compounds' potential for reverse mutation, and the results showed that none induced such mutations. The toxicity prediction also indicated low hERG inhibitory effects. The LD$_{50}$ values, which evaluate acute toxicity, were found to be below 5000 mg/kg, demonstrating optimal safety levels. Most of the compounds fell within this range, classifying them as "Class III" based on predicted acute oral toxicity levels (Table 5). Furthermore, MDS were performed on the top three screened compounds such as beta-amyrin (CID: 73145), episwertenol (CID: 101619548), and kairatenol (CID: 102285188) in comparison to the control drug, ceftobiprole (CID: 135413542). These simulations were conducted based on the compounds' favorable pharmacokinetic profiles and drug-like properties, aiming to evaluate the stability and sustainability of their interactions under physiological conditions typical of the human body. A notable finding in this study is the unique interaction patterns exhibited by these three phytocompounds with the target proteins UspA1 and CEACAM1, highlighting the variability in protein-ligand interactions. However, during MDS, UspA1 demonstrates higher binding stability compared to the human CEACAM1 receptor. This enhanced stability can be attributed to stronger intermolecular interactions and a more favorable conformational landscape in UspA1, allowing it to maintain its binding affinity under varying conditions. The simulation trajectories revealed that UspA1 forms more stable hydrogen bonds and hydrophobic interactions, which contribute to its resilience against fluctuations in temperature and solvent conditions, thereby suggesting its potential as a target for therapeutic interventions against *M. catarrhalis* infections.

The two most important parameters to assess the protein-ligand complex stability used in MDS are RMSD and RMSF. RMSD reflects a compound's maximum stability, whereas RMSF indicates its average fluctuations, providing insights into the protein-ligand complex compactness [58]. In this study, two phytocompounds namely beta-amyrin and kairatenol exhibited significant RMSD and RMSF values when compared to the control drug, indicating their strong interaction with the targeted protein UspA1. Our analysis showed that the therapeutic candidate compounds had consistently lower Rg values compared to the control drug, indicating more compact and stable interactions with the target protein, suggesting enhanced binding affinity and potential therapeutic effectiveness. In protein complexes, a lower Rg reflects tighter and more stable binding between the compound and the protein, whereas a higher Rg suggests

weaker interactions or potential detachment [59]. Furthermore, the SASA values of the three selected compounds also indicated generally favorable characteristics. A lower SASA value suggests that water molecules are closely interacting with the amino acids, leading to a more compact and stable complex. In contrast, a higher SASA value implies that the structure is more exposed and potentially less stable. The SASA values of the three selected compounds examined also indicated generally favorable characteristics [60]. During MDS trajectories, the drug candidates formed a range of interactions with the target protein, including hydrogen bonds, hydrophobic interactions, ionic bonds, and water-bridge bonding. These interactions remained stable throughout the simulation, contributing to strong and persistent binding between the ligands and the protein. Additionally, the MM-GBSA calculations demonstrated that the studied compounds were able to sustain stable and prolonged binding interactions with the target protein UspA1 of *M. catarrhalis*. Numerous studies have shown that *in-silico* predictions of binding free energies align well with experimental results, reinforcing their reliability and predictive accuracy [24,25]. Although experimental binding energy measurements for *M. catarrhalis* UspA1 inhibitors are scarce, comparisons with related bacterial proteins or previous findings on UspA1 inhibitors could offer valuable insights. Research on UspA1 from *Neisseria gonorrhoeae* and *M. catarrhalis* has revealed comparable binding characteristics with specific small molecule inhibitors, offering potential benchmarks for comparison [61,62]. These studies suggest that, despite the limited availability of direct experimental binding data for *M. catarrhalis* UspA1, comparing our results with those from related bacterial systems can help substantiate the *in-silico* findings and reinforce the potential of the tested compounds as effective inhibitors.

Importantly, the MDS trajectories demonstrate that UspA1 forms more stable hydrogen bonds and hydrophobic interactions with beta-amyrin and episwertenol, enhancing the stability of the complexes under varying temperature and solvent conditions. This stability suggests that these compounds could serve as promising targets for therapeutic interventions in *M. catarrhalis*-induced COPD. While binding affinity analysis and interaction profiling indicate that beta-amyrin and episwertenol have strong potential for effective interaction with UspA1 have limitations that should be addressed. One major challenge is the prediction of ADME (absorption, distribution, metabolism, excretion) properties and toxicity profiles without in vitro validation. While computational methods such as molecular docking and molecular dynamics simulations provide insights into binding affinity and protein-ligand interactions, they cannot fully capture the complex pharmacokinetics and off-target effects that occur *in-vivo* [63]. ADME and toxicity predictions often rely on computational models that may not always accurately reflect the biological reality, leading to false positives or negatives in drug discovery [64]. The absence of experimental validation for these predictions can hinder the reliability of the findings. Therefore, while in-silico methods are a promising first step, they must be complemented by in vitro and in vivo studies to confirm the therapeutic potential and safety of the compounds for COPD treatment.

## Conclusions

In summary, our study provides the first evidence that phytocompounds in *S. chirayita* could be a potential alternative treatment for COPD. A chemical library of compounds with structural similarities to *S. chirayita* was created from the IMPPAT 2.0 database, then analyzed and screened for binding energy, drug-likeness, and ADME properties using molecular docking and MDS. Among the 46 phytocompounds screened, beta-amyrin, calendol, episwertenol, kairatenol, and swertanone emerged as potential inhibitors of UspA1 protein from *M. catarrhalis*, showing high binding affinities and stable interactions, including hydrogen bonds and hydrophobic interactions. The ADME properties and toxicity profiles were favorable compared to existing drugs used for bacterial diseases. Beta-amyrin and episwertenol exhibited superior binding stability with UspA1, as confirmed by MDS trajectories. While these

findings are based on *in-silico* analysis, further clinical validation and molecular characterization are needed to confirm their therapeutic potential. These steps are essential to validate the bioinformatic predictions before advancing these phytocompounds for clinical use in treating COPD and other bacterial diseases where UspA1 plays a key role.

## Supporting information

**S1 Fig. The interaction profile of the docking complex reveals the protein-ligands (phytocompounds) interactions.** On upper panel, (A) CEACAM1 and beta-amyrin, (B) CEACAM1 and calendol, (C) CEACAM1 and episwertenol, (D) CEACAM1 and kairatenol, and (E) CEACAM1 and swertanone. The lower panel shows the three-dimensional structure of all complexes.
(JPG)

## Author contributions

**Conceptualization:** Md. Arju Hossain, Md Habibur Rahman, M. Nazmul Hoque.

**Data curation:** Md. Moin Uddin, Md. Shydhur Rahman Chowdhury, Asif Ahsan, Md. Tanvir Hossain, Md. Arif Hossen, Md. Faisal Amin, Rafsan Abir.

**Formal analysis:** Md. Moin Uddin, Md. Shydhur Rahman Chowdhury, Md. Arju Hossain, Asif Ahsan, Abdul Barik, Md. Arif Hossen, Md. Faisal Amin, Rafsan Abir, M. Nazmul Hoque.

**Investigation:** Md. Moin Uddin, Md. Shydhur Rahman Chowdhury, Md. Arju Hossain, Asif Ahsan, Md. Tanvir Hossain, Abdul Barik, Mohammad Shah Alam, Md Habibur Rahman, M. Nazmul Hoque.

**Methodology:** Md. Moin Uddin, Md. Arju Hossain, Asif Ahsan, Md. Tanvir Hossain, Abdul Barik, Md. Arif Hossen, Rafsan Abir, Mohammad Shah Alam, Md Habibur Rahman, M. Nazmul Hoque.

**Project administration:** Md. Arju Hossain, Mohammad Shah Alam, Md Habibur Rahman, M. Nazmul Hoque.

**Resources:** Md Habibur Rahman, M. Nazmul Hoque.

**Software:** Md. Shydhur Rahman Chowdhury, Asif Ahsan, Md. Tanvir Hossain, Abdul Barik, Md. Arif Hossen, Md. Faisal Amin, Rafsan Abir, M. Nazmul Hoque.

**Supervision:** Mohammad Shah Alam, M. Nazmul Hoque.

**Validation:** Md. Arju Hossain, Md. Faisal Amin, Mohammad Shah Alam, Md Habibur Rahman, M. Nazmul Hoque.

**Visualization:** Md. Moin Uddin, Md. Shydhur Rahman Chowdhury, Asif Ahsan, Md. Tanvir Hossain, Abdul Barik, Md. Arif Hossen, Md. Faisal Amin, Rafsan Abir, M. Nazmul Hoque.

**Writing – original draft:** Md. Moin Uddin, Md. Shydhur Rahman Chowdhury, Asif Ahsan, Md. Tanvir Hossain, Abdul Barik, Md. Faisal Amin, Rafsan Abir.

**Writing – review & editing:** Md. Arju Hossain, Mohammad Shah Alam, Md Habibur Rahman, M. Nazmul Hoque.

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
