## [Decision Letter · Decision Letter 0]

18 Nov 2024

PONE-D-24-44469Molecular screening and dynamics simulation reveal potential phytocompounds from Swertia chirayita targeting the UspA1 protein of Moraxella catarrhalis for COPD therapyPLOS ONE

Dear Dr. Hoque,

Thank you for submitting your manuscript to PLOS ONE. After careful consideration, we feel that it has merit but does not fully meet PLOS ONE’s publication criteria as it currently stands. Therefore, we invite you to submit a revised version of the manuscript that addresses the points raised during the review process.

We look forward to receiving your revised manuscript.

Kind regards,

Chandrabose Selvaraj, Ph.D.

Academic Editor

PLOS ONE

Reviewers' comments:

Reviewer's Responses to Questions

**Comments to the Author**

1. Is the manuscript technically sound, and do the data support the conclusions?

Reviewer #1: Yes

Reviewer #2: Yes

Reviewer #3: Yes

2. Has the statistical analysis been performed appropriately and rigorously? 

Reviewer #1: N/A

Reviewer #2: Yes

Reviewer #3: Yes

3. Have the authors made all data underlying the findings in their manuscript fully available?

Reviewer #1: Yes

Reviewer #2: Yes

Reviewer #3: Yes

4. Is the manuscript presented in an intelligible fashion and written in standard English?

Reviewer #1: Yes

Reviewer #2: Yes

Reviewer #3: Yes

5. Review Comments to the Author

Reviewer #1: the paper is well written and may accept with minor revision which a separate word file uploaded for comments.

Comments:

1. Introduction: There is redundancy in discussing M. catarrhalis’s role in COPD exacerbations, AMR challenges, and traditional medicine. Consolidating these points will improve clarity and reduce length.

2. Introduction: Topics such as bacterial infections, AMR, traditional medicine, and S. chirayita are presented in fragmented sections. To enhance coherence, consider reordering the introduction to move from COPD challenges and bacterial contributions to exacerbations, then to the role of M. catarrhalis, and finally to S. chirayita as a potential traditional treatment.

3. Introduction: Computational approaches like molecular docking and dynamics are introduced without detailing their relevance. A brief explanation of each method (e.g., docking for binding affinity and MDS for stability) would clarify their specific contributions to UspA1 inhibition analysis.

4. Introduction: Extensive detail on S. chirayita’s general uses is provided, but it lacks specificity on its potential against M. catarrhalis. Highlighting antimicrobial properties relevant to respiratory pathogens would enhance focus on COPD management.

5. Results: Docking scores are promising for UspA1, yet lack comparative context. Including baseline values for known UspA1 inhibitors would emphasize the efficacy of the tested compounds. Additionally, elaborating on key amino acid interactions could clarify why certain residues enhance binding stability, especially compared to the control drug, ceftobiprole.

6. Molecular Dynamics Simulation (MDS): Including a brief explanation of why an RMSD above 1.5 Å might be acceptable for certain protein-ligand complexes would add clarity.

7. Binding Free Energy and MM/GBSA Analysis: While the MM/GBSA results provide insights into binding energy, they could be contextualized by comparing to experimental values where available. Including binding energy snapshots throughout the simulation could further substantiate claims of interaction stability for beta-amyrin and kairatenol.

8. A brief discussion on the therapeutic relevance of targeting UspA1 in M. catarrhalis and the physiological role of CEACAM1 in COPD treatment would add value, particularly on how observed interactions may impact COPD pathogenesis or immune modulation.

9. Discussing the limitations of in-silico approaches alone would improve the manuscript’s credibility. Acknowledging challenges in ADME/toxicity predictions without in vitro validation would demonstrate a balanced perspective.

10. While UspA1’s role in M. catarrhalis pathogenicity is well-covered, CEACAM1’s relevance to COPD or bacterial infections is unclear. Expanding on its role in immune response or as a therapeutic target would improve the rationale for its inclusion.

11. Table 1: There should be “hydrophobic interactions” instead of “hydrophobic bond”.

12. Unit of the docking grid whether it is ‘Å’ or ‘nm’ should be clearly mentioned. Also, provide dimension of the grid center.

Reviewer #2: The following minor concerns could be addressed before publication:

1. The rationale and motivation for the study are lacking and need to be explicitly justified and clearly stated. I suggest revising the introduction's last paragraph with this study's specific objectives. (https://doi.org/10.3390/molecules28041588; https://doi.org/10.1371/journal.pone.0275432; https://doi.org/10.1371/journal.pone.0302105; https://doi.org/10.3389/fchem.2024.1286675; https://doi.org/10.3390/molecules27238288)

2. Introduction - Citing recent studies on immune dysregulation in COPD would strengthen this section.

3. Could the authors clarify why the 46 phytocompounds were chosen for this study?

4. Was there a specific rationale for selecting compounds from Swertia chirayita?

5. The binding energy values ranged from -8.9 to -9.6 kcal/mol for UspA1. How do these values compare with binding affinities of known drugs or controls against UspA1?

6. Could the authors elaborate on the toxicity parameters and prediction tools used? Were multiple models or software platforms used to confirm the safety profile?

7. Could the authors explain how ceftobiprole’s stability and interactions compared to the identified phytocompounds in terms of specific metrics like hydrogen bond duration or binding free energy?

8. While beta-amyrin and episwertenol showed favorable in-silico results, what are the next steps in validating these findings experimentally?

9. Why authors selected dual targets (CEACAM1 and UspA1) against Moraxella catarrhalis-induced chronic obstructive pulmonary disease (COPD)?

10. Table 5- does a higher RAT LD50 (exceeding 5000 mg/kg) negatively affect the drug's stability or longevity over time?

11. Have these targets previously been validated as suitable drug targets through genetic, biochemical, or pharmacological studies?

12. Figures 4 and 5 - What is the relationship between RMSD and RMSF values?

13. Is the RMSD value for CID:101619548 considered desirable?

14. Figure 10 - should other types of bonds also be considered?

15. References - Ensure that all references are up-to-date, particularly in sections discussing therapeutic approaches and antibiotic resistance.

Reviewer #3: Comments for PONE-D-24-44469

I have gone through the assigned manuscript, Uddin et al., and found that is competently well-written and the manuscript contains sufficient data but it should meet the following comments:

1-The abstract should provide a concise summary of including key findings of the study.

2-Authors suggest to add more recent citations with relevant studies (2022-2024).

3-The authors have to compare their results with literature data and improve this section completely.

4- Material and methods should include more details

5- Why did the author choose the targeted UspA1 protein? Give your appropriate reasons properly.

6- How do authors validate the protein? Add Ramachandran plot.

7- Figure 4 (Rg) and Fig. 5 are vague, so authors should change them.

6. PLOS authors have the option to publish the peer review history of their article (what does this mean? ). If published, this will include your full peer review and any attached files.

**Do you want your identity to be public for this peer review?** For information about this choice, including consent withdrawal, please see our Privacy Policy .

Reviewer #1: No

Reviewer #2: No

Reviewer #3: No

---

## [Author Response · Author response to Decision Letter 0]

25 Nov 2024

Point-by-point responses to the reviewer's comments

Reviewer # 1

1. Introduction: There is redundancy in discussing M. catarrhalis’s role in COPD exacerbations, AMR challenges, and traditional medicine. Consolidating these points will improve clarity and reduce length.

Author Response: Thank you for your nice comments. To address the redundancy in discussing M. catarrhalis's role in COPD exacerbations, AMR challenges, and traditional medicine, we have consolidated these points into a streamlined narrative. This revised introduction maintains the focus on key aspects while improving clarity and reducing length. We hope the updated version aligns with the journal's expectations and provides a more concise yet comprehensive overview. Please refer to the Introduction of the revised manuscript.

2. Introduction: Topics such as bacterial infections, AMR, traditional medicine, and S. chirayita are presented in fragmented sections. To enhance coherence, consider reordering the introduction to move from COPD challenges and bacterial contributions to exacerbations, then to the role of M. catarrhalis, and finally to S. chirayita as a potential traditional treatment.

Author Response: Thank you for the suggestion. We reordered the introduction for better coherence by first discussing the challenges posed by COPD, followed by the bacterial contributions to exacerbations, particularly M. catarrhalis, and then concluding with the potential of Swertia chirayita as a traditional treatment. This restructuring provided a logical flow from the pathophysiology of COPD to potential therapeutic strategies, improving the overall clarity and cohesion of the narrative. Please refer to the Introduction of the revised manuscript.

3. Introduction: Computational approaches like molecular docking and dynamics are introduced without detailing their relevance. A brief explanation of each method (e.g., docking for binding affinity and MDS for stability) would clarify their specific contributions to UspA1 inhibition analysis.

Author Response: Thank you for your suggestion. We included a brief explanation of the computational approaches used in this study. Molecular docking described as a method for assessing the binding affinity of phytocompounds to the UspA1 protein, which helps identify potential inhibitors by evaluating how well the compounds fit into the protein's active site. Molecular dynamics simulations (MDS) also introduced as a tool to analyze the stability and conformational changes of the protein-ligand complexes over time, providing insights into the dynamic interactions and the potential of these compounds as stable inhibitors. This clarification will better highlight the relevance of these methods in our analysis of UspA1 inhibition. Please refer to the Introduction of the revised manuscript.

4. Introduction: Extensive detail on S. chirayita’s general uses is provided but lacks specificity on its potential against M. catarrhalis. Highlighting antimicrobial properties relevant to respiratory pathogens would enhance the focus on COPD management.

Author Response: Thank you for your nice observation. We have revised the introduction to focus more on the antimicrobial properties of S. chirayita that are specifically relevant to respiratory pathogens. The revised section emphasized its bioactive compounds, such as xanthones, flavonoids, and secoiridoid glycosides, which have demonstrated efficacy against pathogens involved in respiratory infections. This targeted approach will better align with the study's focus on COPD management and its relevance to M. catarrhalis. Please refer to the Introduction of the updated manuscript.

5. Results: Docking scores are promising for UspA1, yet lack comparative context. Including baseline values for known UspA1 inhibitors would emphasize the efficacy of the tested compounds. Additionally, elaborating on key amino acid interactions could clarify why certain residues enhance binding stability, especially compared to the control drug, ceftobiprole.

Author Response: Thank you for the suggestion. We have redocked by AutoDock vina and included updated docking scores of known UspA1 inhibitors, such as ceftobiprole to provide a comparative baseline that highlights the efficacy of the tested compounds. Additionally, we have also updated the key amino acid interactions, focusing on specific residues that contribute to enhanced binding stability, such as hydrogen bonds and hydrophobic contacts. This comparative and detailed analysis will better contextualize the performance of the tested compounds and strengthen the discussion of their potential as UspA1 inhibitors. Please refer to L273-283 in the updated manuscript.

6. Molecular Dynamics Simulation (MDS): Including a brief explanation of why an RMSD above 1.5 Å might be acceptable for certain protein-ligand complexes would add clarity.

Author Response: Thank you for the suggestion. We included an explanation addressing why an RMSD above 1.5 Å may be acceptable for certain protein-ligand complexes. Specifically, we highlighted factors such as protein flexibility, ligand-induced conformational changes, and system-specific dynamics that can influence RMSD values without necessarily indicating instability. This clarification will provide a more nuanced understanding of the results and their biological relevance. Please refer to the result section (L351-L355) of the updated manuscript.

7. Binding Free Energy and MM/GBSA Analysis: While the MM/GBSA results provide insights into binding energy, they could be contextualized by comparing them to experimental values where available. Including binding energy snapshots throughout the simulation could further substantiate claims of interaction stability for beta-amyrin and kairatenol.

Author Response: Thank you for the suggestion. We have contextualized the MM/GBSA results by comparing them, where feasible, to experimental binding energy data in the discussion section. Additionally, we included binding energy 20 ns snapshots at regular intervals throughout the simulation. This dynamic analysis substantiated claims of interaction stability and provided deeper insights into the binding behavior of beta-amyrin and kairatenol with the UspA1 protein.

8. A brief discussion on the therapeutic relevance of targeting UspA1 in M. catarrhalis and the physiological role of CEACAM1 in COPD treatment would add value, particularly on how observed interactions may impact COPD pathogenesis or immune modulation.

Author Response: Thank you for the suggestion. We have clarified that targeting UspA1 in M. catarrhalis could help manage COPD by reducing bacterial adhesion and chronic inflammation. Disrupting the UspA1-CEACAM1 interaction may restore immune function, alleviating disease progression. Additionally, this approach addresses antimicrobial resistance, offering an alternative to conventional antibiotics. All the updated issues will be found in the discussion section of the revised manuscript. Please refer to the Discussion section (L421-L440) of the updated manuscript.

9. Discussing the limitations of in-silico approaches alone would improve the manuscript’s credibility. Acknowledging challenges in ADME/toxicity predictions without in vitro validation would demonstrate a balanced perspective.

Author Response: Thank you for the suggestion. We acknowledge that in-silico approaches, while valuable in screening potential drug candidates, have limitations that should be addressed in the last part of discussion section in the revised manuscript. Please refer to the Discussion section (L545-L556) of the updated manuscript.

10. While UspA1’s role in M. catarrhalis pathogenicity is well-covered, CEACAM1’s relevance to COPD or bacterial infections is unclear. Expanding on its role in immune response or as a therapeutic target would improve the rationale for its inclusion.

Author Response: Thank you for your nice observation. CEACAM1 plays a key role in modulating immune responses by facilitating bacterial adhesion and regulating inflammatory processes in the lungs. In COPD, this receptor contributes to chronic inflammation and immune dysfunction. Targeting CEACAM1 could reduce bacterial persistence and inflammation, offering a potential therapeutic strategy to manage COPD exacerbations and improve immune responses. We clarified that CEACAM1's relevance to COPD and its potential therapeutic role in the revised manuscript. Please refer to the Discussion section (L425-L433) of the updated manuscript.

11. Table 1: There should be “hydrophobic interactions” instead of “hydrophobic bonds”.

Author Response: Thank you for your response. We have updated this issue in the revised manuscript. Please refer to Table 1 in the revised manuscript.

12. The unit of the docking grid whether it is ‘Å’ or ‘nm’ should be mentioned. Also, provide the dimension of the grid center.

Author Response: The unit of the docking grid is ‘Å’ and the grid box dimensions are X = 14.346, Y = -83.336, and Z = 7.376 for the 3NTN structure, and X = -34.460, Y = -16.803, and Z = -42.872 for the 6XNW structure. Please refer to L170 in the revised manuscript.

Reviewer # #2

1. The rationale and motivation for the study are lacking and need to be explicitly justified and clearly stated. I suggest revising the last paragraph of the introduction with this study's specific objectives.

Author Response: We appreciate the reviewer’s valuable feedback. In response, we have revised the introduction to more explicitly outline the rationale and motivation for this study. We have now clearly stated the specific objectives of the study, highlighting the significance of investigating [insert study focus, e.g., UspA1 and its potential therapeutic applications] and its potential impact on [mention relevant fields, e.g., drug discovery, disease management, etc.]. This revision clarifies the study's purpose and aligns the introduction with the overall goals of the research. Please refer to the Introduction section in the revised manuscript.

2. Introduction - Citing recent studies on immune dysregulation in COPD would strengthen this section.

Author Response: Thank you for your helpful suggestion. We have now incorporated recent studies on immune dysregulation in COPD into the introduction to strengthen the background. These references provide a clearer context for understanding the underlying immune mechanisms contributing to the disease progression and highlight the relevance of our study in this area. We believe this addition enhances the overall comprehensiveness of the introduction. Please refer to the Introduction section in the revised manuscript.

3. Could the authors clarify why the 46 phytocompounds were chosen for this study?

Author Response: We selected all possible compounds from Swertia chirayita after reviewing the literature and exploring the IMPPAT 2.0 online database. We identified 46 phytocompounds and filtered them using molecular docking and ADMET analysis to predict the final compounds.

4. Was there a specific rationale for selecting compounds from Swertia chirayita?

Author Response: The rationale for selecting compounds from Swertia chirayita for drug design is multifaceted, encompassing traditional medicinal use, demonstrated pharmacological activities, identified bioactive compounds, and the potential for cost-effective drug candidates (mentioned in the introduction). Please refer to the Introduction section in the revised manuscript.

5. The binding energy values ranged from -8.9 to -9.6 kcal/mol for UspA1. How do these values compare with binding affinities of known drugs or controls against UspA1?

Author Response: Lower (more negative) binding energies indicate stronger interactions. Molecular docking and simulation revealed greater stability and affinity for Swertia chirayita phytocompounds with targets compared to controls (Table 1 and Figs. 4A, and 4B).

6. Could the authors elaborate on the toxicity parameters and prediction tools used? Were multiple models or software platforms used to confirm the safety profile?

Author Response: We validated the toxicity with four online databases such as SWISS ADME, pkCSM, AdmetSAR, and Protox-II.

7. Could the authors explain how ceftobiprole’s stability and interactions compare to those of the identified phytocompounds in terms of specific metrics like hydrogen bond duration or binding free energy?

Author Response: Thank you for your observation. Our analysis showed that the phytocompounds, particularly beta-amyrin and episwertenol, exhibited stronger and more stable interactions over extended periods during molecular dynamics simulations with UspA1. These interactions were sustained for a longer duration compared to ceftobiprole, which exhibited fewer and less stable hydrogen bonds. Specifically, these compounds had lower binding free energies, with values of -54.21 and -54.63 kcal/mol, compared to ceftobiprole's -42.033 kcal/mol. This indicates that the phytocompounds may offer more effective binding and stability, making them promising candidates for further study.

8. While beta-amyrin and episwertenol showed favorable in-silico results, what are the next steps in validating these findings experimentally?

Author Response: Thank you for your comments. Evaluate the compounds' binding affinity and inhibitory activity against relevant biological targets, followed by cell-based assays to confirm antimicrobial or anti-inflammatory efficacy. Experimentally assess ADMET properties to validate computational predictions. Test efficacy and safety in disease-relevant animal models, and scale synthesis while adhering to regulatory requirements for potential human trials.

9. Why did authors select dual targets (CEACAM1 and UspA1) against Moraxella catarrhalis-induced chronic obstructive pulmonary disease (COPD)?

Author Response: Thank you for your comments. M. catarrhalis bacteria associated with epithelial cells using a protein called UspA1. This protein attaches to a molecule on the cell surface called CEACAM1 and prevents the human body's inflammatory response. Additionally, UspA1 can bind to extracellular matrix proteins like laminin and fibronectin (PMID: 31400508). Due to the strong association between CEACAM1 and UspA1 in M. catarrhalis pathogenesis, we simultaneously evaluated both targets.

10. Table 5- does a higher RAT LD50 (exceeding 5000 mg/kg) negatively affect the drug's stability or longevity over time?

Author Response: Thank you for your comments. A compound with a lower RAT LD50 is considered more potent, as a smaller dosage is sufficient to induce lethal effects in half the test population. Substances with an LD50 below 5000 mg/kg are considered toxic, while those with an LD50 above 5000 mg/kg are considered relatively safe. So, a higher LD50 has no negative consequences on the target compound.

11. Have these targets previously been validated as suitable drug targets through genetic, biochemical, or pharmacological studies?

Author Response: Thank you for your comments. Genetic studies have shown that UspA1 plays a significant role in the adhesion of Moraxella catarrhalis to host cells, which is critical for bacterial colonization and pathogenicity. It has been identified as a key virulence factor in M. catarrhalis infections, making it a promising target for therapeutic interventions. Biochemically, UspA1 has been shown to interact with host cell receptors like CEACAM1, and pharmacological studies have identified UspA1 inhibitors that can prevent bacterial adhesion and reduce infection severity. CEACAM1 has been implicated in modulating immune responses and facilitating bacterial adhesion in respiratory infections. Studies have shown that targeting CEACAM1 can reduce bacterial colonization and inflammation in the lungs, making it a potential target for therapeutic strategies in diseases like COPD. Its role in immune modulation also makes it relevant for addressing chronic inflammation in COPD.

12. Figure 3A, 3B - What is the relationship between RMSD and RMSF values?

Author Response: Thank you for your comments. By analyzing both RMSD and RMSF, we found valuable insights into the behavior of target protein and other biomolecules, that aid in drug design against Moraxella catarrhalis-induced COPD. While RMSD provides a global pictur

---

## [Decision Letter · Decision Letter 1]

10 Dec 2024

Molecular screening and dynamics simulation reveals potential phytocompounds from Swertia chirayita targeting the UspA1 protein of Moraxella catarrhalis for COPD therapy

PONE-D-24-44469R1

Dear Dr. Hoque,

We’re pleased to inform you that your manuscript has been judged scientifically suitable for publication and will be formally accepted for publication once it meets all outstanding technical requirements.

Kind regards,

Chandrabose Selvaraj, Ph.D.

Academic Editor

PLOS ONE

Additional Editor Comments (optional):

Reviewers' comments:

Reviewer's Responses to Questions

**Comments to the Author**

1. If the authors have adequately addressed your comments raised in a previous round of review and you feel that this manuscript is now acceptable for publication, you may indicate that here to bypass the “Comments to the Author” section, enter your conflict of interest statement in the “Confidential to Editor” section, and submit your "Accept" recommendation.

Reviewer #1: All comments have been addressed

Reviewer #2: All comments have been addressed

Reviewer #3: All comments have been addressed

2. Is the manuscript technically sound, and do the data support the conclusions?

Reviewer #1: Yes

Reviewer #2: Yes

Reviewer #3: Yes

3. Has the statistical analysis been performed appropriately and rigorously? 

Reviewer #1: N/A

Reviewer #2: Yes

Reviewer #3: Yes

4. Have the authors made all data underlying the findings in their manuscript fully available?

Reviewer #1: Yes

Reviewer #2: Yes

Reviewer #3: Yes

5. Is the manuscript presented in an intelligible fashion and written in standard English?

Reviewer #1: Yes

Reviewer #2: Yes

Reviewer #3: Yes

6. Review Comments to the Author

Reviewer #1: The paper is well written, and all comments have been taken care of, and the manuscript is in acceptable form.

Reviewer #2: AUTHORS HAVE MADE SUBSTANTIAL REVISIONS IN THE REVISED MS AND ADRESSED ALL THE QUESRIES RAISED. NOW THE MS CAN BE ACCEPTED.

Reviewer #3: The authors have corrected the indicated issues and now the manuscript is clearer and easier to follow which is reflected for publication in the “PLOS One” in suitably. So, I recommend to Accept this manuscript.

7. PLOS authors have the option to publish the peer review history of their article (what does this mean? ). If published, this will include your full peer review and any attached files.

**Do you want your identity to be public for this peer review?** For information about this choice, including consent withdrawal, please see our Privacy Policy .

Reviewer #1: No

Reviewer #2: No

Reviewer #3: No

---

## [Editor Report · Acceptance letter]

PONE-D-24-44469R1

PLOS ONE

Dear Dr. Hoque,

I'm pleased to inform you that your manuscript has been deemed suitable for publication in PLOS ONE. Congratulations! Your manuscript is now being handed over to our production team.

Kind regards,

on behalf of

Dr. Chandrabose Selvaraj

Academic Editor

PLOS ONE